# Behavioral Effects of Buspirone in Juvenile Zebrafish of Two Different Genetic Backgrounds

**DOI:** 10.3390/toxics10010022

**Published:** 2022-01-07

**Authors:** Amira Abozaid, Robert Gerlai

**Affiliations:** 1Department of Cell and Systems Biology, University of Toronto, Toronto, ON M5S 3G5, Canada; amira.abozaid@mail.utoronto.ca; 2Department of Psychology, University of Toronto Mississauga, Mississauga, ON L5L 1C6, Canada

**Keywords:** anxiety, buspirone, behavioral phenotyping, fear, juvenile zebrafish, strain comparison

## Abstract

Anxiety continues to represent a major unmet medical need. Despite the availability of numerous anxiolytic drugs, a large proportion of patients do not respond well to current pharmacotherapy, or their response diminishes with chronic drug application. To discover novel compounds and to investigate the mode of action of anxiolytic drugs, animal models have been proposed. The zebrafish is a novel animal model in this research. It is particularly appropriate, as it has evolutionarily conserved features, and drug administration can be employed in a non-invasive manner by immersing the fish into the drug solution. The first step in the analysis of anxiolytic drugs with zebrafish is to test reference compounds. Here, we investigate the effects of buspirone hydrochloride, an anxiolytic drug often employed in the human clinic. We utilize two genetically distinct populations of zebrafish, AB_SK_, derived from the quasi-inbred AB strain, and WT, a genetically heterogeneous wild-type population. We placed juvenile (10–13-day, post-fertilization, old) zebrafish singly in petri dishes containing one of four buspirone concentrations (0 mg/L control, 5 mg/L, 20 mg/L or 80 mg/L) for 1 h, with each fish receiving a single exposure to one concentration, a between subject experimental design. Subsequently, we recorded the behavior of the zebrafish for 30 min using video-tracking. Buspirone decreased distance moved, number of immobility episodes and thigmotaxis, and it increased immobility duration and turn angle in a quasi-linear dose dependent but genotype independent manner. Although it is unclear whether these changes represent anxiolysis in zebrafish, the results demonstrate that behavioral analysis of juvenile zebrafish may be a sensitive and simple way to quantify the effects of human anxiolytic drugs.

## 1. Introduction

Anxiety represents a large unmet medical need as despite decades of research and the development of numerous anxiolytic drugs, a large number of patients suffer from this disorder [1]. Part of the reason for this is that we do not understand the mechanisms underlying anxiety. Furthermore, anxiety is not a homogeneous disease from a mechanistic standpoint, but rather a collection of different disorders with distinct etiologies and potentially diverse underlying mechanisms [2,3]. Briefly, the molecular and neurobiological mechanisms and genetic pathways underlying anxiety remain to be understood, e.g., [3,4,5,6].

To tackle the above complex questions, well-controlled laboratory research using a variety of animal models has been proposed. The majority of these studies employ traditional laboratory research organisms, i.e., the house mouse and the rat [7,8,9,10,11,12,13,14,15,16,17]. Nevertheless, more recently, novel model organisms have also been proposed. The zebrafish is one such model organism [18,19,20,21,22,23,24]. This latter species is gaining popularity in psychopharmacology and neuroscience research for numerous reasons. It is prolific and easy to house in large numbers in small vivaria. It is amenable to high-throughput screening and has been successfully employed in mutation as well as drug screens [20,25,26]. The zebrafish exhibits several evolutionarily conserved features, including high (on average about 70%) nucleotide sequence homology of its genes to that of mammalian, including human, orthologs [27,28,29]. Another advantage of the zebrafish is the non-invasive and simple drug administration possible with this species. The drugs can be effectively administered to the fish by immersing them into the drug solution at controlled concentrations and for the desired duration of time [21,30]. Additionally, the effects of mutations and drugs can be efficiently detected by the growing number of behavioral test paradigms developed for this species [19,31,32,33,34,35].

Anxiety and fear responses are particularly well characterized in zebrafish [22]. Some argue that anxiety and fear are distinct phenomena in mammals with potentially different underlying mechanisms [36,37]. The former is the lasting behavioral state or prolonged response to diffuse aversive stimuli or contexts and is often thought of as a chronic pathological problem. The latter is believed to be an acute response to clearly present immediate danger and is often thought to be adaptive. In zebrafish, the manifestations, i.e., the phenotypical features, of fear and anxiety are usually not distinguished, and thus here we will refer to all aversive reactions of zebrafish as anxiety-like responses.

Numerous aversive stimuli and test contexts have been successfully employed to trigger anxiety-like responses in zebrafish. Ranging from novelty of the test tank [18,19,20] to precisely timed delivery of aversive computer animated visual stimuli [38,39], these aversive cues have been shown to elicit a rich repertoire of anxiety-like responses. These responses include reduction of activity (often measured as distance travelled or swim speed), elevated immobility, elevated erratic movement, often measured as increased turn angle, and increased thigmotaxis or “wall-hugging” [22]. In nature, such responses are believed to be adaptive, as they may allow the fish to escape from predators. For example, reduced swimming activity or increased time spent completely immobile allows the prey to escape detection by visually hunting predators or from those that perceive lateral line cues (vibrations in the water). Increased erratic movement or zig-zagging, which is often observed near or on the bottom of experimental tanks [40], may disturb the loose detritus that accumulates on the bottom of slowly moving streams or lakes, creating a cloud of dust surrounding the prey fish [41]. Increased thigmotaxis may also be effective in nature as proximity to physical objects (the wall of the tank in the lab or the side of the trunk of an immersed tree in nature) reduces the area of attack by, and makes the prey fish less detectable to, the approaching predator [42]. Indeed, these behavioral measures have been successfully employed to quantify anxiety-like responses and changes in such responses induced in psychopharmacology analyses [43,44,45,46]. We note, however, that anxiety-like responses have been found to represent a rich behavioral repertoire, and how exactly zebrafish respond depends upon the environmental context as well as the nature of the aversive stimulus [39]. This complexity may explain the occasional controversies surrounding how and what to measure as an anxiety-like response in zebrafish [47].

Buspirone hydrochloride is an anxiolytic drug prescribed to patients diagnosed with general anxiety disorder or GAD [48]. Along with benzodiazepines, including diazepam, clorazepate, alprazolam and lorazepam, it has been found effective for GAD [48]. However, unlike benzodiazepines, buspirone use does not induce sedation, physical dependence, or cognitive and psychomotor impairment. However, other side effects including dizziness, headaches, nausea, nervousness, and paresthesia have been reported among patients taking buspirone [48]. In mammals, buspirone is known to have high affinity to the 5HT1A receptor, acting on both at the pre- and post-synaptic terminals. It is an agonist whose main action is believed to be through the 5HT1A pre-synaptic autoreceptor, leading to inhibition of serotoninergic neurotransmission [48]. In addition to its low affinity interactions with other 5HT receptors, buspirone is also known to be an antagonist for the dopamine D2 autoreceptor in mammals, which inhibits dopaminergic neurotransmission [48]. In zebrafish, its psychopharmacological profile or binding characteristics/specificity have not been systematically analyzed.

Nevertheless, multiple model organisms, including zebrafish, have been tested for their response to buspirone [21,35,49,50,51,52,53,54,55,56,57,58,59,60,61,62]. In adult zebrafish, buspirone has been found to decrease the diving response [63], i.e., bottom-dwelling, without altering swim speed in a novel tank task, an effect considered to be anxiolytic [21]. Similarly, buspirone has been found to decrease duration of immobility, a change that is consistent with anxiolysis [49]. In a black and white choice paradigm, buspirone increased the time zebrafish spent in the white zones, i.e., led to decreased scototaxis, [35,49] again interpreted as proof for anxiolytic properties of buspirone in zebrafish. Interestingly, neither social cohesion nor locomotion were found altered by buspirone in adult zebrafish despite that these behaviors have been shown to be affected by aversive contexts and stimuli [53].

A possible reason why anxiolytic drugs affect patients differently is their genetic makeup. In animal models too, strain, i.e., genetic origin-based, differences in drug effects have been demonstrated. For example, strain dependent effects of benzodiazepines have been shown in rodents [64,65], including in the magnitude of sedation, anxiolytic actions [66,67], as well as in the development of physical dependence [68]. Buspirone has also been found to exert its effects in a strain dependent manner [69].

In our current study, we will investigate the effects of buspirone hydrochloride in juvenile zebrafish (10–13 days old post-fertilization) of two genetically distinct zebrafish populations. The effects of this drug on juvenile zebrafish have not been investigated. The advantage of using zebrafish of this young age is that the fish are small (about 7 mm long at this age), and thus require only small amount of space, which makes them amenable to large scale high throughput screens using standard multi-well plates [70]. Furthermore, unlike for adults that would take four months to develop and grow, these young fish can be employed less than two weeks after fertilization.

Here, we report significant and dose-dependent behavioral changes induced by buspirone that were independent of the genotype (population origin) of the juvenile zebrafish we tested. Thus, although at this proof-of-concept stage we cannot ascertain whether the behavioral alterations induced by buspirone represent reduced anxiety, our results demonstrate that juvenile zebrafish may be an appropriate tool with which brain-function-altering effects of human anxiolytic drugs may be detected and screened efficiently.

## 2. Materials and Methods

### 2.1. Animals and Housing

All experiments described with zebrafish in this study have been approved by the Local Animal Care Committee of the University of Toronto Missisauga (protocols # 20011872, Approval Code: 00004716, Approval Date: 20 February 2020). Adult zebrafish were obtained from a local pet store (Big Al’s Aquarium Warehouse, Mississauga, ON, Canada). We call the descendants of these fish “wild type” (WT). We decided to employ pet store fish for breeding the WT population in order to maximize genetic variability, i.e., increased heterozygosity ratio for the loci of genes for each individual and increased genetic variability in the WT population as a whole. Because Big Al’s Aquarium obtains their fish from commercial breeding facilities where the number of breeding pairs is large, we assume that the decreased level of inbreeding meets the above criteria. We have also argued that the increased genetic variability and high heterozygosity ratio makes the WT population more representative of species-typical features of zebrafish found in nature [44,71]. Unlike this genetically heterogeneous population, standard zebrafish strains may have unique, strain-specific idiosyncratic features. Numerous such quasi-inbred zebrafish strains have been developed, the AB strain being, perhaps, one of the most frequently used among them. The second population of zebrafish that we employed in this study originated from this strain. We call these fish ABSK because they came from the Hospital for Sick Children (Sick Kids (SK) Zebrafish Facility, Toronto, ON, Canada). In this study, we refer to these two groups of zebrafish as populations or genotypes. We note that although we have not ascertained genetic differences using genetic markers, these two populations have been separated by multiple decades of independent breeding and, thereby, are expected to be genetically distinct, simply due to genetic drift.

Adults of both populations were housed in 2.8 L tanks on Aquaneering, Inc. (San Diego, CA, USA) system racks. They were fed Zeigler zebrafish micro-pellets (Zeigler Bros, Inc., Gardners, PA, USA) twice a day, as well as brine shrimp (A. salina, Brine Shrimp Direct, Ogden, UT, USA) once daily. Five days prior to breeding, the fish were separated by sex and placed into similar-sized tanks (2.8 L) on system racks. Two males and two females were then placed in breeding tanks. Four breeding tanks per genotype were used in total. The breeding tank had a perforated bottom to allow for collecting eggs, after spawning has occurred. Both WT and ABsk zebrafish were placed in breeding tanks at 5:00 p.m. to allow spawning to take place the next morning. The automated light-cycle system at the zebrafish facility mimics dawn, where lights slowly turn on at 6:30 a.m. and full lighting is achieved an hour later by 7:30 a.m. The full lighting phase usually marks the start of spawning. After spawning, the parental fish were returned to their home tanks, and eggs were collected at 9:00 a.m. The eggs were washed using sterilized and system water (conductivity = 150 μS, pH = 7.0). Thirty fertilized eggs were placed in each petri-dish (9 cm diameter) filled with sterile system water. The water was kept at 27 °C. Unfertilized eggs were removed daily, and 10% of the water was replaced with fresh sterile system water every day. Zebrafish larvae reached free swimming stage at 6 days post-fertilization (dpf). From this time onward, the zebrafish larvae were fed twice a day with API 100 (Zeigler^®^, Gardners, PA, USA) powdered dry fry food. At their age of 10 dpf juvenile zebrafish were moved to 9 L tanks on system racks filled with system water of identical conductivity, pH and temperature as used before. The fish were used for behavioral testing at their age of 10–13 dpf. We call these fish “juveniles”, and not “larval”, as we argue zebrafish of this developmental stage do not undergo rapid and dramatic tissue reorganization characteristic of metamorphosis.

### 2.2. Experimental Design and Behavioral Testing

We employed a 4 × 2 between subject experimental design with buspirone concentration having 4 levels 0 mg/L (control), 5 mg/L, 20 mg/L, or 80 mg/L) and genotype having 2 levels (WT or ABSK population origin). The buspirone concentrations were selected based upon prior studies with zebrafish [21,49,50,53] and also upon our own preliminary studies, showing no side effects, mortality or morbidity even at the highest dose chosen. The drug solutions were prepared in the laboratory using buspirone hydrochloride tablets. Whole buspirone tablets with a known drug amount per tablet were crushed and homogenized in system water. The drug solution was then filtered using coffee paper to remove any solid vehicle or binding agent used in the tablet. Twenty fish were tested per concentration and population, i.e., a total of 160 juvenile zebrafish were tested. All fish were tested only once in a manner randomized across concentration and genotype, so that, on average, fish from each of the eight groups were tested around the same time.

The juvenile zebrafish were fed API 100 (Zeigler^®^, Gardners, PA, USA) fry food an hour prior to exposure to buspirone and subsequent to behavioral testing. The experimental subject was singly pipetted into a 35 mm diameter petri-dish holding the corresponding drug solution (9 mL total volume). The duration of exposure of the juvenile fish to buspirone was 60 min. Although detailed absorption, distribution, metabolism and excretion (ADME) characterization of compounds is lacking in zebrafish, several psychopharmacology studies used 30 min acute exposure periods in adult zebrafish (for examples see [28]), and some used as short as 3 min [21]. In juvenile/larval zebrafish, shorter than 30 min acute exposure periods (larger body surface/volume ratio, hence faster absorption) may be sufficient. Nevertheless, we decided to be conservative, and employed 60 min exposure length, because in our hands too within 30 min most drugs, including buspirone, exerted their maximum level behavioral effects, i.e., the effect plateaued. Subsequent to acute buspirone administration fish were singly pipetted into an identically sized petri- dish filled with the same drug solution the subject was exposed to before the behavioral testing. Both exposure and behavioral testing petri-dishes were placed on a Picker light-table; thus, the petri dishes were illuminated from below. The level of illumination in the buspirone exposure and behavioral recording petri-dish was found to be 1800 Lux (Lux Light Meter Pro v2 developed by Elena Polyanskaya for the Apple Iphone). This high level of illumination accomplished two things. One, it created an aversive environment. Open and well-illuminated areas without hiding places potentially expose the fish to predators and have been shown to be aversive for zebrafish [72,73]. Two, the high illumination level enhanced our ability to track the movement of the small subject. Two video cameras were positioned 30 cm above the petri-dishes (JVC GZ-MG330HV). Each camera viewed four petri-dishes. Thus, fish of all eight groups could be recorded and tested simultaneously. The behavioral trials were 30 min long each. The recordings were transferred from the camera to external hard drives, and later replayed and analyzed using Ethovision XT 13 (Noldus Info Tech., Wageningen, The Netherlands) video-tracking software.

Numerous behavioral parameters of the swim path of the fish were quantified that previously were shown to be associated with anxiety-like responses and/or exploratory behavior of adult zebrafish in novel environments [31,74]. We measured the total distance the fish moved (cm). General swimming activity has been shown to decrease under aversive conditions. Absolute turn angle (degree) measures the amount of turning irrespective of direction. The direction of swimming may change for a number of reasons in zebrafish but turn angle has been shown to correlate well with erratic movement (or zig-zagging), a typical anti-predatory reaction shown to be exhibited by zebrafish under aversive conditions or in response to fear-inducing stimuli. Duration of immobility (s) has also been found to positively correlate with fear or anxiety and is believed to be an effective anti-predatory response allowing the small zebrafish to remain undetected to predators. Immobility here is defined as fewer than 20% of the number of pixels corresponding to the total visible surface area of the subject changing from one video-frame to the next (with 30 frame per s temporal resolution). Although intuitively total distance travelled and immobility duration may seem to be redundant, negatively correlating measures, empirical data have suggested that they represent independent behavioral responses [41,44,71,75]. Frequency of immobility (number of immobility episodes) was also quantified as it may represent a behavioral strategy independent of the duration of immobility. Finally, the duration of time within the thigmotaxis zone (s) and the frequency (number) of entries to this zone were also measured. The thigmotaxis zone was the perimeter of the petri-dish, defined as a 5 mm thick annulus from the side wall of the petri-dish.

### 2.3. Statistical Analysis

SPSS (version 24) written for the PC (windows 10) was used for statistical analysis. Repeated measures variance analyses (ANOVAs) were performed to test the effects of Interval (the within subject repeated measure factor with 30 levels, the number of 1 min intervals of the behavioral recording session), Buspirone hydrochloride concentration (4 levels, the number of concentrations of buspirone employed) and Population with 2 levels (the two genotypes of fish measured) and the interactions among these factors. Population (genotype) effects were found non-significant for all behaviors. Additionally, Interval effects were found non-significant, or the performance of fish was found to show random temporal changes (no clear time-dependent trend) for all behaviors. Furthermore, these factors were found not to interact with buspirone effects. For these reasons, and because repeated measures designs are inappropriate for post hoc multiple comparison tests, we pooled the data across intervals as well as across the two populations, and further investigated the effect of Buspirone using univariate ANOVA (4 drug concentrations). To compare buspirone concentration effects without committing type−1 error, the univariate ANOVA was followed by the post hoc multiple range Tukey Honestly Significant Difference (HSD) test. Significance in all our statistical tests was accepted when the probability of the null hypothesis (no effect or no difference) was found not larger than 5%, i.e., when *p* ≤ 0.05.

## 3. Results

Confirming the results of our preliminary studies, at the buspirone concentrations and with the dosing method employed, we did not observe any mortality, morbidity or gross behavioral abnormalities. Nevertheless, quantification of the swim path parameters of buspirone-treated zebrafish did reveal notable changes. For example, the total distance the fish swam appeared affected by buspirone treatment similarly in the two populations of juvenile zebrafish tested (Figure 1A,B). Repeated measures ANOVA confirmed this observation and found a significant Interval F(29, 4234) = 5.400, *p* < 0.001) effect, a non-significant Interval × Population (F(29, 4234) = 0.298, *p* > 0.99), Interval × Buspirone (F(87, 4234) = 1.048, *p* > 0.35) and Interval × Strain × Buspirone (F(87, 4234) = 0.929, *p* > 0.65) interaction. The two populations were also not found to differ significantly from each other (F(1, 146) = 0.342, *p* > 0.55). However, the effect of Buspirone was found to be highly significant (F(3, 146) = 9.252 *p* < 0.001), while the Population × Buspirone interaction was non-significant (F(3, 146) = 0.226, *p* > 0.85). Analysis of the data pooled across interval and population showed a linear dose dependent effect of buspirone, with the highest dose reducing total distance most (Figure 1C). ANOVA confirmed the buspirone effect was significant (F(1, 3) = 9.364, *p* < 0.001). Tukey HSD post hoc test showed that the 0 mg/L concentration group significantly differed from the 20 mg/L (*p* = 0.009) and the 80 mg/L concentration groups (*p* < 0.000), and also that the 5 mg/L concentration group significantly differed from the 80 mg/L concentration group (*p* < 0.01).

Analysis of the duration of immobility showed a different effect profile, although here too buspirone was found to affect both genotypes similarly (Figure 2A,B). Repeated measures ANOVA revealed a significant Interval effect (F(29, 4234) = 2.294, *p* < 0.001), a significant Interval × Population interaction (F(29, 4234) = 1.524, *p* < 0.05), but found the Interval × Buspirone (F(87, 4234) = 1.088, *p* > 0.25) and the Interval × Strain × Buspirone interactions (F(87, 4234) = 0.864, *p* > 0.80) non-significant. The Population difference was found significant F(1, 146) = 5.966, *p* < 0.05) and the effect of Buspirone was found highly significant (F(3, 146) = 7.909, *p* < 0.001). However, the Population × Buspirone interaction was non-significant (F(3, 146) = 0.293, *p* > 0.83). Univariate ANOVA conducted for the pooled data confirmed the highly significant Buspirone effect (F(1, 3) = 7.776, *p* < 0.001) (also see Figure 2C). Tukey HSD post hoc tests showed that fish of the 0 mg/L buspirone group (*p* = 0.001) and fish of the 5 mg/L buspirone group (*p* = 0.001) significantly differed from fish of the 80 mg/L group.

The number of times the juvenile zebrafish stopped moving (immobility frequency) also appeared to be significantly affected by buspirone treatment in a concentration dependent manner (Figure 3). Repeated measures ANOVA found the effect of Interval significant (F(29, 4234) = 3.329, *p* < 0.001), but the Interval × Population (F(29, 4234) = 0.65, *p* > 0.95), Interval × Buspirone (F(87, 4234) = 0.863, *p* > 0.80) and Interval × Population × Buspirone interactions (F(87, 4234) = 1.172, *p* > 0.10) were non-significant. The main effects of Population (F(1, 146) = 5.816, *p* < 0.05) and Buspirone (F(3, 146) = 14.721, *p* < 0.001) were significant, but the Population × Buspirone interaction (F(3, 146) = 0.293, *p* > 0.83) was not. ANOVA of the pooled data confirmed the highly significant Buspirone effect (F(1, 3) = 14.095, *p* < 0.001). Post hoc Tukey HSD test revealed that fish of the 0 mg/L and of the 5 mg/L groups differed from fish of the 20 mg/L and 80 mg/L groups (*p* < 0.001).

The Absolute turn angle was also affected by buspirone treatment in a concentration dependent manner (Figure 4). ANOVA found a significant Interval effect (F(29, 4234) = 3.682, *p* < 0.001). However, the Interval × Population (F(29, 4234) = 1.096, *p* > 0.30), Interval × Buspirone (F(87, 4234) = 0.946, *p* > 0.60) and Interval × Population × Buspirone (F(87, 4234) = 0.834, *p* > 0.85) interactions were non-significant. The effect of Population did not reach the level of significance (F(1, 146) = 3.548, *p* = 0.062). However, the effect of Buspirone was found highly significant F(3, 146) = 18.336, *p* < 0.001). The Population × Buspirone interaction was non-significant (F(3, 146) = 0478, *p* > 0.65). ANOVA of the pooled data showed that indeed Buspirone had a highly significant effect (F(1, 3) = 18.051, *p* < 0.001), and Tukey HSD revealed that fish of the 0 mg/L and 5 mg/L groups differed from fish of the 20 mg/L (*p* < 0.01) and 80 mg/L groups (*p* < 0.001).

The number of entries to the thigmotaxis zone showed a large amount of variation but buspirone appeared to reduce it in a dose dependent manner (Figure 5). Repeated measures ANOVA found the effect of Interval to be significant (F(29, 4234) = 1.803, *p* < 0.01), but the Interval × Population (F(29, 4234) = 1.013, *p* > 0.40), Interval × Buspirone F(87, 4234) = 0.894, *p* > 0.70) and Interval × Population × Buspirone (F(87, 4234) = 0.916, *p* > 0.65) interactions were non-significant. The Population effect was also non-significant (F(1, 146) = 2.323, *p* > 0.10). However, the effect of Buspirone was found highly significant (F(3, 146) = 19.555, *p* < 0.001), while the Population × Buspirone interaction was not (F(3, 146) = 0.757, *p* > 0.50). ANOVA of the pooled data confirmed the significant buspirone effect (F(1, 3) = 18.811, *p* < 0.001), and Tukey HSD test showed that fish of the 0 mg/L group differed from fish of the 20 and 80 mg/L groups (*p* < 0.001) and also that fish of the 5 mg/L group differed from fish of the 20 mg/L (*p* < 0.05) and 80 mg/L (*p* < 0.001) groups.

## 4. Discussion

In this study, we have analyzed the effects of buspirone, an anxiolytic drug commonly prescribed for human generalized anxiety disorders, on the behavior of 10–13 dpf old juvenile zebrafish in a novel well-illuminated test arena. Buspirone treatment was found not to have resulted in increased mortality or morbidity or in gross impairment in the ability to move. However, when we quantified the swim-path parameters of zebrafish of two distinct genetic backgrounds, populations, with the use of video-tracking, we did detect buspirone-induced changes. Our results demonstrated significant and dose dependent acute buspirone effects that were independent of the population origin of the experimental zebrafish. The buspirone effects were also consistent across the entire 30 min observation period, as demonstrated by the lack of significant interaction between the main factors of Interval (time) and Buspirone concentration.

Although the main effect of Interval was found significant for all behaviors analyzed, this effect only reflected minute to minute fluctuations rather than a consistent direction of change with time. This result is expected because by the time zebrafish were placed into the test arena, they had been swimming in their respective buspirone bath for an hour, and the same buspirone concentration was continued during the subsequent 30 min of behavioral recording. Pharmacodynamic aspects of buspirone, including drug absorption, distribution, metabolism and excretion (ADME) related processes, have not been characterized for zebrafish, including larval or juvenile zebrafish. Nevertheless, we expected the one-hour long bath immersion to lead to stable drug levels in the brain of the subjects. This is because even in the case of slow absorption or fast metabolism, the large volume of the bath and the constant exposure to the selected buspirone concentration are expected to lead to a steady equilibrium between the external (bath) and internal (brain) drug concentration. In other words, bath immersion is expected to clamp the concentration of buspirone close to that of the external bath. We also note that most psychopharmacology studies employed with zebrafish utilized shorter, some of them much shorter, bath immersion periods, and still found significant drug effects [21,53]. Several of these studies tested adult zebrafish in which the body surface-to-volume ratio was smaller than in the small juvenile zebrafish we used in the current study. The speed of absorption is expected to positively correlate with body surface-to-volume ratio; thus, we speculate that absorption should be faster in our juvenile zebrafish. In summary, although we did not measure it, we expected that the concentration of buspirone would reach a steady maximum level in equilibrium with that of the external bath in the brain of our juvenile zebrafish after the 60 min pre-test bath immersion. We also expected this internal concentration to remain stable throughout the subsequent 30 min exposure, i.e., during the behavioral test period. The lack of consistent temporal changes during this latter period is in accordance with this expectation.

The most important results of this study concern the behavioral changes acute buspirone treatment induced. Fish exposed to buspirone reduced their distance moved (swam less), stayed immobile longer but less frequently. They also turned more, and visited the thigmotaxis zone less frequently. Importantly, these buspirone effects were found linearly dose dependent for the behavioral parameters distance moved, immobility frequency and number of entries to the thigmotaxis zone, with the strongest effects seen in fish exposed to the highest (80 mg/L bath) buspirone concentration. Immobility duration and turn angle also showed such dose dependency, albeit less perfect, i.e., with fish of the lowest dose group (5 mg/L) statistically indistinguishable from the 0 mg/L control fish. The quasi-linear dose dependency of acute buspirone effects thereby demonstrates that the 10–13 dpf old juvenile zebrafish is an excellent subject with which drug effects may be demonstrated efficiently.

The finding of no interaction between genotype (population origin) and buspirone concentration demonstrates that buspirone-affected fish of the two population similarly. Whether this finding can be extended to several other genotypes, including quasi-inbred standard strains or other genetically heterogeneous random bred populations of zebrafish is difficult to predict, but it certainly shows that buspirone’s effects were not modulated by genetic differences. Therefore, we conclude that the small, easy and fast to produce 10–13 dpf old zebrafish juvenile may be an excellent laboratory organism in which effects of drugs may be detected. We also emphasize that the simple behavioral analysis, video-tracking-based quantification of swim-path parameters in an open arena may allow researchers to detect the effects of a variety of toxicants, compounds, small molecules or drugs that are absorbed from the exposure bath.

The last question we consider is whether the behavioral changes buspirone induced can be interpreted as reduced anxiety-like responses. This is not a simple question to address for two main reasons. One, the zebrafish has been shown to possess a rich anti-predatory (fear/anxiety response) behavioral repertoire that is suspected to be context and aversive stimulus specific [72,76]. In other words, depending on numerous features of the environment and depending on the type and strength of the aversive (fear or anxiety inducing) stimulus, zebrafish may respond differently. In fear and anxiety research, and in studies of anti-predatory behavior, scientists often distinguish two major classes of responses, active and passive avoidance reactions. However, the response repertoire of most species, including that of the zebrafish, is not unidimensional, i.e., varies not just along the linear scale of activity-passivity. The second reason why it may be difficult to ascertain whether the buspirone-induced behavioral changes represent anxiolysis is that anxiety and fear-related or anti-predatory responses of juvenile zebrafish are not well understood. Some of the changes we found in response to acute buspirone exposure appear to be inconsistent with anxiolysis. The two examples of this are distance moved and duration of immobility. Distance moved, swim speed, or general activity level is expected to drop under aversive conditions in zebrafish [38,74]; thus, anxiolytic drugs are expected to have activity increasing effects. Here, however, we found a linear dose dependent reduction of distance moved in fish to whom buspirone was administered. The literature, however, does provide some controversies as swimming activity has also been found to increase, rather than decrease, in response to fear inducing stimuli in some zebrafish studies [71,77]. Immobility has also been used to quantify effects of fear and anxiety inducing stimuli or drugs. Immobility has been shown to increase under aversive conditions and anxiolytic drugs are expected to reduce this response. Here, however, we found the opposite: buspirone increased the duration of immobility. Buspirone does not have a known sedative effect, at least in mammals [78,79,80]; thus, these above two changes in activity levels are likely due to other behavioral mechanisms. It is possible that juvenile zebrafish, unlike adults, respond to danger with active rather than passive avoidance reactions. If this is correct, the decreased swimming activity and increased duration of immobility induced by buspirone could reflect reduction of an active avoidance reaction via the anxiolytic effect of buspirone. However, it is also possible that these buspirone-induced changes have nothing to do with anxiety, and instead are due to motor dysfunction [81], a question that will be addressed by empirical studies in the future.

The changes induced by buspirone treatment in the remaining three behavioral measures are consistent with anxiolytic effects. Buspirone was found to reduce the number of occasions the treated zebrafish stopped moving. Frequency of immobility has been found to increase under aversive conditions and decrease in response to anxiolytic drug treatment [18,82]. Similarly, turn angle has been found to increase in response to fear or anxiety inducing stimuli [31,38,82] and decrease in response to anxiolytic drugs [83,84], a change we found induced by buspirone in the current study. Finally, increased thigmotaxis, or wall hugging, has been shown to be a sign of anxiety in a variety of species, from rodents [85,86,87,88,89,90] to zebrafish [42,46,77,91,92,93]. In response to anxiolytic drugs, thigmotaxis has been shown to diminish, a finding consistent with what we observed in our buspirone exposed juvenile zebrafish. Nevertheless, as anxiety-like responses are less understood in juvenile or larval zebrafish, as mentioned before, other possibilities may also need to be considered, including motor-function-altering effects of buspirone. The latter possibility is supported by finding serotoninergic neurons in the zebrafish spinal cord [94], demonstrated involvement of the serotoninergic neurotransmitter system in zebrafish locomotion [95], and significant motor impairment found after fluoxetine (a selective serotonin reuptake inhibitor) administration in larval zebrafish [96].

In summary, although the precise interpretation of acute buspirone administration induced behavioral changes will require thorough psychopharmacological characterization of the juvenile zebrafish, this current proof-of-concept study demonstrated that the juvenile zebrafish can be an efficient and cost-effective tool with which brain-function-altering effects of compounds or small molecules may be tested. These molecules can be drug candidates. The small size of the 10–13 dpf old zebrafish along with the simplicity of behavioral testing represents a highly scalable system, a prerequisite for high throughput screening. Thus, we suggest that these small fish will be useful for large scale compound library screens. The molecules to be tested can also be environmental toxicants or other agents of toxicology studies too. The simplicity and non-invasive nature of compound administration via the bath immersion along with the efficiency of the behavioral analysis also suggests good utility of these small fish in such studies.

## Figures and Tables

**Figure 1 toxics-10-00022-f001:**
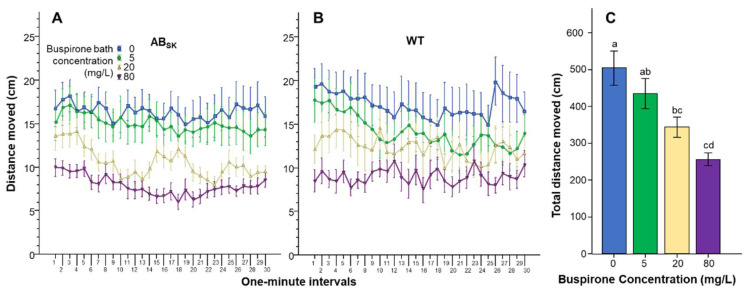
The distance fish moved is dose dependently reduced by buspirone. Distance as a function of time (1-min intervals) is shown for ABSK (**A**) and WT (**B**) zebrafish. The legends show the buspirone concentrations employed (blue line with rectangle 0 mg/L, green line with circle 5 mg/L, yellow line with triangle 20 mg/L, and purple line with upside down triangle 80 mg/L). (**C**) shows the cumulative (total) distance moved across the entire 30 min recording session, pooled for the two populations, by the zebrafish exposed to the different concentrations of buspirone (indicated underneath the X axis and also by the same color coding as in (**A**,**B**). Means + S.E.M. are shown. The results of Tukey HSD test are shown as letters above each bar of the bar graph (**C**). Bars that do not share one letter in common represent buspirone treatment groups that are significantly (*p* < 0.05) different from each other. For detailed results of statistical analyses, see Results.

**Figure 2 toxics-10-00022-f002:**
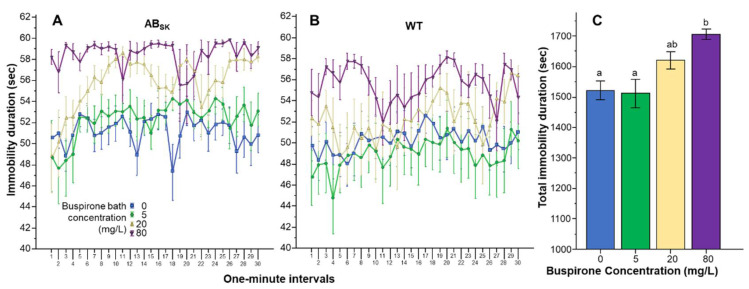
Duration of immobility is dose dependently increased by buspirone. Immobility duration as a function of time (1-min intervals) is shown for ABSK (**A**) and WT (**B**) zebrafish. The legends show the buspirone concentrations employed (blue line with rectangle 0 mg/L, green line with circle 5 mg/L, yellow line with triangle 20 mg/L, and purple line with upside down triangle 80 mg/L). (**C**) shows the cumulative (total) duration of immobility for the entire 30 min recording session, pooled for the two populations, exhibited by the zebrafish exposed to the different concentrations of buspirone (indicated underneath the X axis and also by the same color coding as in (**A**,**B**)). Means + S.E.M. are shown. The results of Tukey HSD test are shown as letters above each bar of the bar graph (**C**). Bars that do not share one letter in common represent buspirone treatment groups that are significantly (*p* < 0.05) different from each other. For detailed results of statistical analyses, see Results.

**Figure 3 toxics-10-00022-f003:**
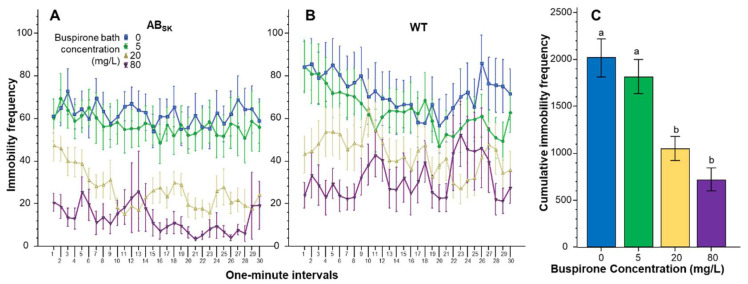
The number of times fish stopped moving (frequency of immobility) is dose dependently reduced by buspirone. Immobility frequency as a function of time (1-min intervals) is shown for ABSK (**A**) and WT (**B**) zebrafish. The legends show the buspirone concentrations employed (blue line with rectangle 0 mg/L, green line with circle 5 mg/L, yellow line with triangle 20 mg/L, and purple line with upside down triangle 80 mg/L). (**C**) shows the cumulative (total) number of immobility episodes for the entire 30 min recording session, pooled for the two populations, exhibited by the zebrafish exposed to the different concentrations of buspirone (indicated underneath the X axis and also by the same color coding as in (**A**,**B**)). Means + S.E.M. are shown. The results of Tukey HSD test are shown as letters above each bar of the bar graph (**C**). Bars that do not share one letter in common represent buspirone treatment groups that are significantly (*p* < 0.05) different from each other. For detailed results of statistical analyses, see Results.

**Figure 4 toxics-10-00022-f004:**
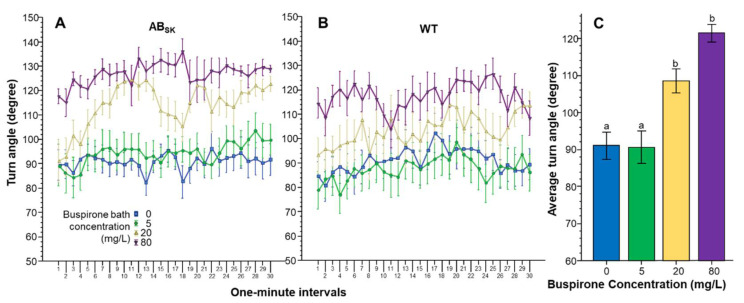
Turn angle is dose dependently increased by buspirone. Turn angle as a function of time (1-min intervals) is shown for ABSK (**A**) and WT (**B**) zebrafish. The legends show the buspirone concentrations employed (blue line with rectangle 0 mg/L, green line with circle 5 mg/L, yellow line with triangle 20 mg/L, and purple line with upside down triangle 80 mg/L). (**C**) shows the average turn angle calculated for the entire 30 min recording session, pooled for the two populations, exhibited by the zebrafish exposed to the different concentrations of buspirone (indicated underneath the X axis and also by the same color coding as in (**A**,**B**)). Means + S.E.M. are shown. The results of Tukey HSD test are shown as letters above each bar of the bar graph (**C**). Bars that do not share one letter in common represent buspirone treatment groups that are significantly (*p* < 0.05) different from each other. For detailed results of statistical analyses, see Results.

**Figure 5 toxics-10-00022-f005:**
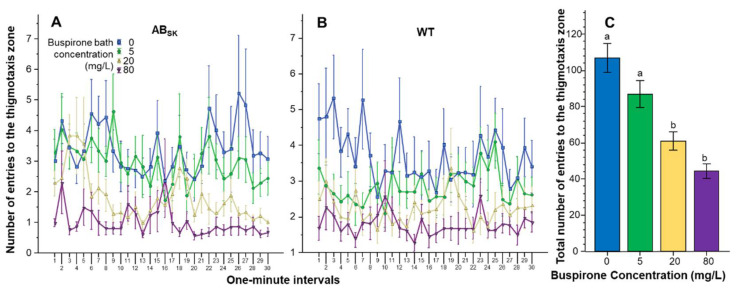
The number of entries to the thigmotaxis zone is dose dependently reduced by buspirone. Thigmotaxis zone entry frequency as a function of time (1-min intervals) is shown for ABSK (**A**) and WT (**B**) zebrafish. The legends show the buspirone concentrations employed (blue line with rectangle 0 mg/L, green line with circle 5 mg/L, yellow line with triangle 20 mg/L, and purple line with upside down triangle 80 mg/L). (**C**) shows the cumulative (total) number of entries to the thigmotaxis zone for the entire 30 min recording session, pooled for the two populations, exhibited by the zebrafish exposed to the different concentrations of buspirone (indicated underneath the X axis and also by the same color coding as in (**A**,**B**). Means + S.E.M. are shown. The results of Tukey HSD test are shown as letters above each bar of the bar graph (**C**). Bars that do not share one letter in common represent buspirone treatment groups that are significantly (*p* < 0.05) different from each other. For detailed results of statistical analyses, see Results.

## Data Availability

Raw data obtained in this study are available from the authors upon request.

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
